# Chemo-Enzymatic Generation of Highly Fluorescent Nucleoside Analogs Using Purine-Nucleoside Phosphorylase

**DOI:** 10.3390/biom14060701

**Published:** 2024-06-14

**Authors:** Alicja Stachelska-Wierzchowska, Jacek Wierzchowski

**Affiliations:** Department of Physics and Biophysics, University of Warmia and Mazury in Olsztyn, 10-719 Olsztyn, Poland; alicja.stachelska@uwm.edu.pl

**Keywords:** purine nucleoside phosphorylase, nucleoside analogs, fluorescence, chemo-enzymatic synthesis

## Abstract

Chemo-enzymatic syntheses of strongly fluorescent nucleoside analogs, potentially applicable in analytical biochemistry and cell biology are reviewed. The syntheses and properties of fluorescent ribofuranosides of several purine, 8-azapurine, and etheno-purine derivatives, obtained using various types of purine nucleoside phosphorylase (PNP) as catalysts, as well as α-ribose-1-phosphate (r1P) as a second substrate, are described. In several instances, the ribosylation sites are different to the canonical purine N9. Some of the obtained ribosides show fluorescence yields close to 100%. Possible applications of the new analogs include assays of PNP, nucleoside hydrolases, and other enzyme activities both in vitro and within living cells using fluorescence microscopy.

## 1. Introduction: Fluorescent Nucleoside Analogs—An Outlook

Fluorescence spectroscopy is one of the most sensitive methods of biomolecules detection and measurements of their interactions and activities, at both the ensemble and single-molecule levels [1,2,3]. For this reason, various fluorescence techniques are frequently used in biological, biochemical, and biophysical research. Unfortunately, some of the most important biomolecules (nucleic acids, membranes, some proteins) are not fluorescent under natural conditions (neutral aqueous medium, moderated temperatures). Therefore, the direct application of fluorescence methods to biological research in vivo or in vitro requires finding appropriate fluorescent markers, which can provide structural and dynamic information about molecules as well as their activities. 

Among these, fluorescent nucleoside and nucleotide analogs play key roles [4]. They are applied, inter alia, to the structural and dynamic research of nucleic acids, including ribozymes [5,6,7,8], their interaction with proteins [9], and the detection of enzymatic activities of enzymes related to nucleic acid metabolism both in vitro and in vivo with the aid of fluorescence microscopy [10,11,12]. 

Natural nucleosides and nucleotides are virtually nonfluorescent in neutral aqueous media [13] with the exception of some weakly fluorescent minor RNA subunits [14]. The first fluorescent nucleoside analogs, among them the intensely fluorescent 2-aminopurine riboside, were described five decades ago [15], and some of them are still employed in biophysical research [16,17]. Since their development, hundreds of fluorescent nucleobase and nucleoside analogs have been developed for various applications, most of them by multi-step chemical procedures [4,6,10,18,19,20,21].

There are two broad classes of nucleoside analogs [5]: (i) nucleosides linked to strong aromatic or heteroaromatic fluorophores, like coumarin, pyrene, or fluorene, typically via the C-8, N7, or ribose moiety; and (ii) nucleosides with a heteroaromatic ring (purine or pyrimidine) replaced by a fluorescent analog of nucleobase [5,21]. The most useful analogs are those that combine good spectral parameters with substrate or inhibitory activities towards the selected enzymatic or other biological system, and this condition is usually fulfilled by isosteric or isomorphic (that is, mimicking the natural nucleosides in size, shape, and hydrogen-bonding abilities) analogs. In this brief review, we concentrate mainly on this kind of nucleoside analog.

There is a growing number of such analogs that are known, and recent findings of Tor’s group enlarged the available “libraries” by the substituted thieno- and thiazolo-pyrimidine compounds, which are isosteric and to a large extent isofunctional with natural purine nucleosides [22]. All the thieno- and isothiazolo-pyrimidine nucleoside analogs are strongly fluorescent in the visible region [22,23]. These are all C-nucleosides, which makes them resistant to some nucleolytic enzymes, which is important for biological applications.

Many nucleoside analogs, including elements of the “RNA emissive alphabet” can be selectively incorporated into polynucleotides and ribozymes, either chemically or by using enzymes like T7 RNA polymerase and other polymerases [5,22]. 

Most fluorescent analogs are generated chemically, using sometimes quite complex multi-step procedures, but some can be obtained in much simpler, enzymatic ways [24,25]. Purine-nucleoside phosphorylase (see below) offers the possibility of nucleoside (but not of the C-nucleoside) analogs’ synthesis from nucleobase and pentose analogs.

There are also examples of fluorescent pyrimidine nucleoside analogs [5,6,21,26,27], but nearly all of them belong to the first (“noncanonical”) group, with pyrimidine moieties being linked to an organic fluorophore, or significantly modified. They are nevertheless utilized in polynucleotide and nucleic acid research, e.g., for the detection of apurinic sites in double-stranded polynucleotides or DNA fragments [28].

## 2. Purine Nucleoside Phosphorylase as a Synthetic Tool

Purine nucleoside phosphorylase (PNP, E.C. 2.4.2.1) catalyzes a reversible phosphorolysis of natural purine nucleosides (or deoxynucleosides) to the respective nucleobases and α-(deoxy)ribose-1-phosphate ((d)r1P [29]). It is a key enzyme of the so-called purine salvage pathway [30], and its physiological role is the regulation of nucleoside concentrations within the cells. 

A genetic deficiency of PNP activity leads to the severe impairment of the immune system in humans [29,31], and it is a subject of experimental gene therapy [31]. Inhibitors of PNP reveal potent pharmacological activities towards bacterial, viral, and parasitic infections (including malaria) and can also be applied as antineoplastic and immunosuppressive agents [29,32]. One of them, the transition-state inhibitor forodesine (known as immucilin H [33]), with an inhibition constant of ~60 pM, has recently been approved in Japan for the treatment of relapsed/refractory peripheral T-cell lymphoma [34]. The even more potent inhibitor ulodesine [35], with K_i_ ~9 pM, is currently under examination.

Various forms of PNP are employed as catalysts in the enzymatic syntheses of nucleosides and their analogs (for reviews, see Refs. [24,25,36,37,38,39]). For canonical purines and many analogs, the reaction equilibrium is shifted towards nucleoside synthesis [29], but there are also ribosides for which phosphorolysis is virtually irreversible, and these are ideal to serve as ribose donors in the synthetic “one pot” reactions, while purified r1P is rarely used because of its high price and relative instability [36]. A commonly known example is the commercially available 7-methylguanosine [40], and other good ribose donors include 7-methyl-thioguanosine, 7-methyl-inosine, and nicotinamide riboside [29]. Another method of “one-pot” purine (deoxy)nucleoside synthesis was proposed long ago by Krenitsky [41], with uridine or thymidine phosphorylase, acting on uridine or thymidine, serving as a source of r1P or its deoxy analog.

More than 100 natural forms of PNP are known [38]. Various forms of PNP, both natural and modified via the site-directed mutagenesis [39], differ quantitatively or qualitatively in substrate specificities and may be used for synthetic purposes. There are two broad classes of PNP: (i) mammalian (trimeric) forms which are specific towards 6-oxopurine nucleosides [29], but a single genetic mutation in the active site, N243D, can alter this specificity [42]; and (ii) bacterial forms, which are mostly hexameric and reveal a broader substrate specificity, particularly with respect to the heteroaromatic moieties of the substrates. Thermophilic PNP variants [43] offer opportunities for large-scale applications. Whole-cell catalysis is also recognized for its relative simplicity and enzyme stability [36,39]. Various advanced techniques, including those utilizing enzyme immobilization on magnetic epoxide microspheres, have led to effective syntheses on a gram or even kilogram scale [44]. Analysis of the industrial potential of such techniques has recently been presented [39].

PNP, particularly its bacterial forms, is characterized by broad substrate specificity, and this refers both to the heteroaromatic and sugar moieties, as well to various non-typical arrangements of these [38]. Early works pointed unexpected activities of mammalian PNP against N3- and N7-ribosides of canonical purines [45], as well as effective modifications of enzyme specificities via site-directed mutagenesis of the active-site amino acids [42]. These observations led to many applications in the syntheses of chemically, biochemically, and pharmacologically important compounds, like arabinosyl-purines, cordycepin (3’-deoxyribosyl-adenine) and analogs, halogenated purine and benzimidazole ribosides and deoxyribosides, including cladribine (2-chlorodeoxyadenosine), and fludarabine (2-fluoro-arabinosyl-adenine) as well as many other nucleoside analogs, reviewed recently [37]. All known PNP forms maintain strict regioselectivity in the ribosylation site (purine N9) and pentose configuration (β); rare exceptions to the former rule are discussed below. Attempts to further modify enzyme specificity in a controlled manner are underway [39,43,46].

Other enzymatic systems affecting deoxynucleoside production are deoxyribosyl transferases (DNT’s) of bacterial origin [36,47]. The analogous syntheses of nucleotides from nucleobases are possible using phosphoribosyl transferases [48].

In the present paper, we summarize recent progress in obtaining new highly fluorescent nucleoside analogs and point out their potential for analytical applications. This paper is an update of our previous, more extended review [49].

## 3. Chemo-Enzymatic Synthesis of Fluorescent Ribosides of Purines and 8-Azapurines

The only strongly fluorescent purine riboside is 2-aminopurine riboside, but this compound is apparently not ribosylated by PNP. Its deoxy analog can be synthesized using DNT as a catalyst [36]. Weak fluorescence is exhibited by 2,6-diaminopurine riboside and N^6^,N^6^-dimethyladenosine [49], and these analogs, and the corresponding (deoxy)ribosides, can be synthesized chemo-enzymatically using either PNP from *E. coli* or bacterial DNTs as catalysts [36]. Moderately fluorescent 7-methylguanosine and 7-methylinosine undergo almost irreversible phosphorpolysis by both trimeric and hexameric PNPs [29], so they cannot be synthesized in this way. In contrast to the “canonic” purines, the 8-azapurines show much better spectral properties and are substrates for many forms of PNP in the synthetic pathway [49,50].

The 8-azapurines are isosteric and isomorphic analogs of natural purine bases (see Figure 1), substituting the latter in a variety of biochemical transformations [49,50,51,52,53]. Some of them are quite toxic, and to date no direct medical application of these compounds has been reported, but they are widely employed in laboratories for various purposes, including mutagenicity tests [51,52]. Their chemical and physicochemical properties differ somewhat from those of the analogous purines, particularly in the acido-basic equilibria and prototropic tautomerism [52]. Nevertheless, they can be classified as probably the most isomorphic purine analogs, although their emissive properties are inferior to those of the “RNA alphabet”. 

In contrast to the canonical nucleobases, some of the 8-azapurines, as well as the corresponding nucleosides and nucleotides, reveal marked fluorescence in neutral aqueous medium (see Refs. [49,54] and Table 1). Their nucleotides can be selectively incorporated into mRNA molecules, including ribozymes, and have been applied to mechanistic research of ribozyme action [55,56].

### 3.1. 8-Azaadenosine, 8-Azainosine, 8-Azaxanthosine and Derivatives

Although 8-azaadenine is very weakly fluorescent, and only as a neutral species (pH < 6.1), 8-azaadenosine exhibits moderate fluorescence in a broad pH range [50]. Incorporated into ribozyme molecules, it serves as a reporter of the acid-base equilibria important for the reaction mechanism [55]. As expected, wild-type PNP from *E. coli* accepts 8-azaadenine as a substrate, but the ribosylation proceeds about 50-fold slower than that for adenine [50]. Mutations of bacterial PNP thus far have not been effective in improving the rate of ribosylation.

The compound 8-azainosine (synthesized chemically) was investigated by Seela and co-workers [53], who reported moderate fluorescence of the anionic form of the nucleoside (pK_a_ ~8). The compound 8-azahypoxanthine reveals poor substrate properties towards PNP in the synthetic pathway, since in the neutral aqueous medium it exists as an anion [53]. 

The compound 8-azaxanthine (8-azaXan) and particularly its methyl derivatives are known to be strongly fluorescent in aqueous medium [53,57], but the chemo-enzymatic synthesis of 8-azaxanthosine from 8-azaxanthine is difficult, probably because this compound is even more acidic than most azapurines (pK_a_ ~4.8, [51]). It has been reported, however, that the inducible xanthosine phosphorylase from *E. coli*, known as PNP-II [58], accepts 8-azaxanthine as a substrate in the ribosylation pathway [57]. The ribosylation site is certainly not N9, but either N7 or N8, as inferred from the spectral properties of the ribosylation product(s), which are strongly fluorescent at 440 nm in the neutral aqueous medium, similarly to N7- or N8-methyl, but not to N9-methyl derivatives of 8-azaXan [53].

### 3.2. 8-Azaguanosine and Analogs

The compound 8-azaguanine is known as a very toxic analog of guanine, which is applied in mutagenicity tests [59]. It reveals quite intense fluorescence as a neutral species (<pH 7), ascribed to its minor N(8)H tautomer [50]. By contrast, 8-azaguanosine is very weakly fluorescent below pH 6, where it exists in the neutral form, but strongly as a monoanion (pK_a_~8). This compound is readily, and almost irreversibly, synthesized enzymatically from 8-azaguanine, using calf spleen PNP as a catalyst, and r1P as a ribose donor [50]. It has been applied to mechanistic research of ribozyme action, primarily as a probe of micro-acidity of the active guanine residue of the *glms* ribozyme [56].

The mutated form of calf PNP, N243D, in an analogous (slower) reaction gave a mixture of N9- and moderately fluorescent N-7 riboside [60]. By contrast, the application of the bacterial (*E. coli*) enzyme led (rather slowly) mainly to N9- riboside, with some minor contribution of another strongly fluorescent product, tentatively identified as N8-riboside [60,61].

The compound 8-azaguanine forms a weak but fluorescent complex with calf spleen PNP, which was a subject of spectroscopic investigations [62]. The fluorescence excitation spectrum of the complex was markedly shifted to the long wavelength relative to analogous spectrum of the free 8-azaGua. Interestingly, the weakly fluorescent N9-phosphonomethoxyethyl analog of 8-azaguanine [63] binds to this enzyme quite tightly as a bi-substrate analog, also showing a marked increase in fluorescence. Somewhat surprisingly, these two complexes are spectrally quite different [62], the fact interpreted in terms of the prototropic tautomerism of the bound 8-azaguanine, possibly involving the enol tautomer.

The compound 8-aza-deoxyguanosine has been also synthesized, on a larger scale, using PNP as a catalyst [64], and its fluorescence is reported to be very similar to that of the riboside [65].

### 3.3. Other 8-Azapurine Analogs and Their Enzymatic Ribosylation

Although 2,6-diaminopurine is only moderately fluorescent, and its riboside even less [15], their 8-aza analog 2,6-diamino-8-azapurine (DaaPu) reveals a high fluorescence yield in neutral aqueous media [53]. Even stronger emission is observed in some ribosides of DaaPu (Table 1), which could be generated enzymatically using mammalian (calf) or bacterial (*E. coli*) forms of PNP [61]. The ribosylation site depends in this case qualitatively on the catalyst used. Wild-type calf PNP ribosylates DaaPu at N7 and N8, while its mutated form (D243N) ribosylates DaaPu at N7 and N9, and the bacterial (*E. coli*) enzyme at N9 and N8 [61]. All three ribosides differ markedly by their spectroscopic parameters, particularly fluorescence maxima (spectral data summarized in Table 1). The N9-riboside of DaaPu is probably the most intensely fluorescent nucleoside analog known to date, with a yield reaching 90% (Table 1).

The non-typical ribosides of DaaPu, after purification, revealed good and selective substrate properties towards human erythrocytic and bacterial (*E. coli*) forms of PNP, allowing the quantitation of PNP activity in whole human blood, diluted 1000-fold, without the necessity of the removal of hemoglobin [66]. Mammalian and bacterial forms of PNP can be detected selectively, since the N7-riboside of DaaPu is resistant to the *E. coli* PNP, and the N8-riboside is apparently not phosphorolysed by human erythrocytic enzyme [66]. This assay is possible because of marked spectral differences between the above-mentioned non-typical ribosides and the nucleobase analog (DaaPu). This selectivity in PNP ribosylation was a subject of molecular modeling investigations [67], which can lead to further broadening of the synthetic possibilities.

The effect of non-typical ribosylation of DaaPu may be kinetic in nature, at least in the case of *E. coli* PNP, as evidenced by the bi-phasic nature of the reaction progress. Initially, probably a kinetically produced mixture of N9- and N8-ribosides is eventually slowly converted to the final thermodynamic product, which is the “canonical” N9-riboside. The reaction progress is easily observed using fluorescence spectroscopy, thanks to marked differences between N9- and N8-ribosides, emitting at 370 nm and 430 nm, respectively (see Appendix A, Figure A1). This is, to our knowledge, probably the first reported exception from the generally postulated thermodynamic control of the reactions catalyzed by PNP [68].

It may be also interesting that the phosphonomethoxyethyl analog of DaaPu, PME-DaaPu, prepared by Holy et al. [63,69], is also intensely fluorescent in the near UV region, and as a bi-substrate analog strongly binds to the mammalian PNP in the absence of phosphate ions [49].

The compound 8-aza-deoxy-isoguanosine has been obtained chemically by Seela and Jiang [70] and polymerized to study oligonucleoside associations. This deoxyriboside is strongly fluorescent, particularly in the anionic form [70]. Unlike many other nucleosides, 8-aza-isoguanosine ribosides are also fluorescent in polymers [70,71], and may serve as reporters of their structure and interactions. The chemo-enzymatic synthesis of 8-aza-isoguanosine is possible but difficult because of the marked acidity of the corresponding purine analog (pK ~5.5, [53]), so at pH > 6 only a small fraction of molecules is in the neutral form. Preliminary experiments (unpublished) conducted in acetate buffer at pH 5.5, at 30 ℃, showed a slow reaction occurring, leading to a moderately fluorescent ribosylated product, but the ribosylation site remains problematic.

### 3.4. 8-Aza-7-Deazapurine and 8-Aza-9-Deazapurine Ribosides

The title compounds (known also as pyrazolo[3,4-d]pyrimidine and pyrazolo[4,3-d]pyrimidine ribosides, respectively) are isomorphic purine analogs, revealing strong biological and pharmaceutical activities, exemplified by allopurinol (4-oxo-pyrazolo[3,4-d]pyrimidine), an inhibitor of xanthine oxidase, widely applied in anti-gout therapy [72]. The corresponding ribosides and deoxyribosides can be synthesized enzymatically using bacterial (*E. coli*) PNP [64]. This reaction is much slower than that for natural purines, apparently because protonation of the purine N7 is essential for the substrate activation by the D204 residue [53]. According to Mikhailopulo, this role is in this case taken by the Ser90 residue of the *E. coli* PNP, interacting with the purine N8 [73]. 

Some of the pyrazolo[3,4-d]pyrimidine ribosides are fluorescent, as indicated by the fluorescence of the commercial nucleobase analog 8-aza-7-deaza-isoguanine (4-amino-6-hydroxyprazolo[3,4-d]pyrimidine), which is a slow substrate of PNP from *E. coli* (unpublished). Fluorescence changes, observed during the reaction (unpublished) show that at least one of the products must be fluorescent.

The ribosides of 8-aza-7-deaza-2-aminopurine and 8-aza-7-deaza-2,6- -diaminopurine, (IUPAC names: 6-aminopyrazolo[3,4-d]pyrimidine and 4,6-diaminopyrazolo[3,4-d]pyrimidine, respectively) were prepared and modified chemically by Seela et al. [74] and enzymatically by Mikhailopulo’s group [64]. Both compounds exhibit measurable fluorescence, which was applied in the studies of polynucleotide interactions [74].

The 8-aza-9-deazapurine ribosides, known as formycins [75], are C-nucleosides and as such are not phosphorolysed by PNP, but reveal good substrate properties towards other enzymes of purine metabolism and some of them, as well as the respective nucleotides, are fluorescent in neutral aqueous media [15,76,77]. These compounds strongly inhibit hexameric PNPs [29] and their fluorescence was utilized in investigations of enzyme–inhibitor complexes, especially to identify tautomeric forms of the bound ligand(s) [78,79].

### 3.5. Other Bicyclic Heteroaromatics

Benzimidazole fluorescence has been known from many years, and this compound and many derivatives are readily (deoxy)ribosylated by PNP from *E. coli* [24]. Also, halogenated derivatives of benzimidazole have been reported to be substrates of PNP [24,80,81]. Another fluorescent isomer, indazole (benzopyrazole), is probably a poor substrate or not a substrate. Benzoxazole is apparently not a substrate for the PNP from *E. coli* [24], but enzymatic ribosylation of 2-aminobenzoxazole is possible, and apparently takes place on the amine group [73]. N1-deaza and N3-deaza purines can be ribosylated by bacterial forms of PNP, but their fluorescence is not known [73].

## 4. Chemo-Enzymatic Synthesis and Properties of The Tricyclic Purine Analogs and Their Ribosides

Tricyclic purine analogs, although they cannot be classified as isomorphic with the parent purines, typically combine good spectral characteristics with substrate activities towards many enzymes as well as biological activities [82,83,84]. The simplest way to obtain such analogs is chloroacetaldehyde treatment of amino-group-containing purines, the reaction which can be performed in aqueous medium and in relatively mild conditions [85]. Some of the “etheno” derivatives (Figure 2) are good substrates for various forms of PNP in the synthetic pathway, and their ribosylation leads to strongly fluorescent ribosides, as shown below.

Much more difficult is the synthesis of the “extended” nucleobase and nucleoside analogs, containing an additional phenyl ring between the pyrimidine and imidazole moieties [83]. These “lin-benzo” and other “extended nucleosides” reveal strong fluorescence in the visible range and substrate activities towards some important enzymes of purine metabolism (e.g., adenosine and guanosine deaminase) [83]. The chemistry and biological activity of the tricyclic purine analogs have been summarized in recent reviews [85,86].

### 4.1. Adenosine Analogs

The strongly fluorescent 1,N^6^-ethenoadenosine (εAdo) is easily generated chemically from the parent adenosine by treatment with chloroacetaldehyde in aqueous medium at room temperature [82]. Although not susceptible to phosphorolysis by calf PNP, it can be readily phosphorolysed to 1,N^6^-ethenoadenine (εAde, **I,** see Figure 2) by the bacterial (*E. coli*) enzyme [87]. This latter reaction is reversible, but in the synthetic pathway, HPLC analysis revealed a minor contribution of the adenine ribosylated at N7. By contrast, the ribosylation of the etheno-adenine base by calf PNP as a catalyst and r1P as a ribose source quite effectively provides a non-typical and moderately fluorescent N^6^-riboside. The mutated form of calf PNP (N242D) in the analogous reaction provided a mixture of N^6^- and N9- ribosides [87]. The N^6^-riboside is only moderately fluorescent (Table 2) but can be useful for the detection of PNP activity in biological or clinical material, including diluted whole blood, since it is rapidly phosphorolysed by the human erythrocytic PNP to strongly fluorescent 1,N^6^-ethenoadenine [66].

### 4.2. Guanosine Analogs

Two isomeric ethenoguanosine derivatives (**II** and **III**, Figure 2) are generated from the parent guanosine by treatment with chloroacetaldehyde in aqueous medium [88]. This reaction is slower than that of adenosine, so several different strategies were proposed to achieve this goal [89]. One of the etheno-guanine isomers, 1,N^2^-ethenoguanine, is a good substrate for bacterial (*E. coli*) PNP, but not for the calf PNP, providing a poorly fluorescent nucleoside (Table 2). The second isomer, N^2^,3-ethenoguanine, is nonfluorescent and resistant to ribosylation by PNP [90].

Somewhat stronger fluorescence (yield 0.11, see Table 2) was detected from N^2^,3-etheno-O^6^-methylguanosine [90], which can be enzymatically generated from the base analog by the *E. coli* PNP, but in this case the ribosylation rate was ~300-fold slower relative to guanosine synthesis.

### 4.3. 2-Aminopurine Riboside and Isoguanosine Analogs

The reaction of chloroacetaldehyde with the commercial 2-aminopurine riboside is rapid and gives essentially one product, identified as 1,N^2^-etheno-2-aminopurine-N9-β-D-riboside [91]. This compound is only moderately fluorescent, but the analogous reaction with 2-aminopurine base yielded two products, identified as 1,N^2^-etheno-2-aminopurine (**IV**, ~95%) and N^2^,3-etheno-2-aminopurine (**V**, ~5%), the latter showing a fluorescence yield >70% [92]. The minor product (**V**) was found to be a good substrate of both calf PNP and *E. coli* PNP, giving fluorescent riboside products [92]. A much slower reaction was observed for the “linear” 1,N^2^-etheno-2-aminopurine (**IV**) with both enzymes. Somewhat unexpectedly, the ribosylation takes place on N^2^ nitrogen, rather than the “canonical” N9 (Figure 3 and Ref. [92]). The purified riboside of N^2^,3-etheno-2-aminopurine riboside is rapidly phosphorolysed by calf and human erythrocytic PNP, and the reaction is easily followed fluorometrically thanks to the marked spectral difference between the substrate and the product [92]. Applications of these compounds may be somewhat uncomfortable because of their sensitivity to air oxidation. Another possible application of the “linear” etheno-2-aminopurine derivative may be in the fluorometric detection of xanthine oxidase activity (in preparation).

Isoguanine (2-hydroxy-adenine), which is an almost nonfluorescent product of adenine radiolysis [93], reacts with chloroacetaldehyde quite rapidly, giving essentially one product, identified as 1,N^6^-etheno-isoguanine or 1,N^6^-etheno-2-oxo-adenine [90], moderately fluorescent in neutral aqueous medium (Table 2). This compound is readily ribosylated by the *E. coli* PNP, and more slowly by the calf enzyme [90], with r1P as a ribose donor. There are at least three products of such ribosylation reactions, all of them fluorescent, with the fluorescence quantum yield reaching 0.66 for the compound identified as 1,N^6^-etheno-2-oxo-adenine-6-β-D-riboside (see Table 2), and this was the main product when the reaction was catalyzed by calf PNP [90]. 

Attempts to obtain highly fluorescent nucleobase analogs from 2,6-diaminopurine riboside reacting with chloracetaldehyde were not successful [94], but our preliminary data indicate that some of the related ribosides were in fact intensely fluorescent.

### 4.4. Tricyclic Azapurine Analogs

Enzymatic phosphorolysis of the moderately fluorescent 2-aza analog of εAdo, prepared originally by Tsou et al. [95], has been reported [49], but the reverse reaction has not been examined.

Some other tricyclic 2-aza-purine, 8-aza-purine, and 2,8-diazapurine nucleoside analogs were synthesized chemically, with the results summarized in Ref. [96]. The reported fluorescence quantum yields are lower than those obtained for purine analogs, but the reported spectra are batochromically shifted relative to those of εAdo [96]. To our knowledge, no data on enzymatic hydrolysis of phosphorolysis of these analogs exist.

**Table 2 biomolecules-14-00701-t002:** Fluorescence parameters of selected “etheno” substituted purine ribosides in water. Unless otherwise indicated, all data refer to neutral species of the fluorophores. Data from [82,87,90,92].

Compound	Excitation: λ_max_ [nm]	Emission: λ_max_ [nm]	Quantum Yield	Decay Time [ns]
1,N^6^-etheno-adenosine	305	410	0.56	21
1,N^6^-etheno-6-β-D-ribosyl-adenine	310	380	0.10	nd
1,N^2^-ethenoguanosine (anion)		400	<0.01	nd
N^2^,3-ethenoguanosine	261	400	0.02	nd
N^2^,3-etheno-O^6^-methylguanosine	272	405	0.11	nd
1,N2-etheno-2-aminopurine	248	473	0.18	6.9;10.3
1,N^2^-etheno-2-aminopurine-9-β-D- riboside	295	463	0.14	nd
1,N^2^-etheno-2-aminopurine-2-β-D- riboside	338	~405	~0.7	nd
N^2^,3-etheno-2-aminopurine	315	406	0.73	3.8;8.5
N^2^,3-etheno-2-aminopurine-2-β-D- riboside	315	357	0.29	2.15
1, N^6^-etheno-2-hydroxy-adenine	295	415	0.17	nd
1, N^6^-etheno-2-hydroxy-adenosine	295	415	0.34	6.1
1,N^6^-etheno-2-hydroxy-adenine-7-β-D- riboside	294	360	0.036	0.8;5.2
1,N^6^-etheno-2-hydroxy-adenine-6-β-D- riboside	303	425	0.66	nd

nd = no data.

## 5. Conclusions and Perspectives

This brief review illustrates the possibility of the rapid and effective syntheses of many strongly fluorescent ribosides, which can be used in future biophysical and analytical research. The presented examples include various forms of PNP used as catalysis and illustrate the possibilities of modifying enzyme specificity via site-directed mutagenesis not only in a quantitative sense but sometimes in a qualitative sense. Variability in the ribosylation sites exhibited by PNP in some reactions is helpful for the preparation of ribosides with better spectral characteristics, suitable, e.g., for fluorescence microscopy. This variability was not observed for canonical purines, with the only exception of the recently reported synthesis of xanthine-N7-β-D-riboside [97]. The recent paper of Narczyk et al. [98] suggesting the functional non-equivalence of PNP subunits for the *E. coli* PNP may provide a partial explanation of this phenomenon.

Efforts to modify the specificity of PNP using various genetic modifications have been published [99]. Recent achievements in this field involve the chemo-enzymatic syntheses of antivirals islatravir and molnupiravir, synthetized using enzymatic cascades, with a significant contribution by PNP, engineered by “artificial evolution” [100,101]. In the future, specific enzymes may be designed for specific substrates to be ribosylated at specific sites, and theoretical methods potentially leading to this goal are being developed [102]. Sensitive and specific fluorometric assays for PNP, briefly presented above, can be regarded as “bonuses” of this kind of research. 

It would be interesting to examine analogous possibilities of other enzymatic systems, for example, phosphoribosyl transferases, catalyzing the formation of nucleotides from nucleobases and 5-phosphoribose-1-pyrophosphate (PrPP), a group of enzymes which are extensively investigated because of their medical significance [103,104,105]. Various bacterial enzymatic systems, like deoxy-ribosyl transferases, phosphoribosyl transferases, and/or nucleoside hydrolases (acting in non-aqueous media) may also be useful, but their specificities towards fluorescent or fluorogenic analogs are mostly unknown.

## Figures and Tables

**Figure 1 biomolecules-14-00701-f001:**
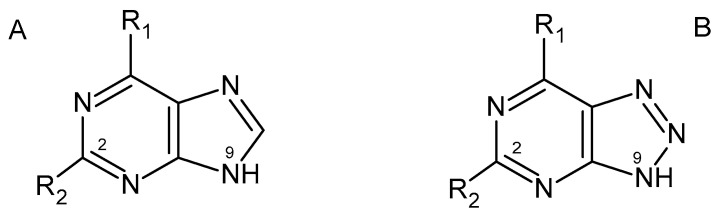
Showing structural similarity between purines (**A**) and 8-azapurines (**B**). Note that purine numbering is maintained throughout.

**Figure 2 biomolecules-14-00701-f002:**
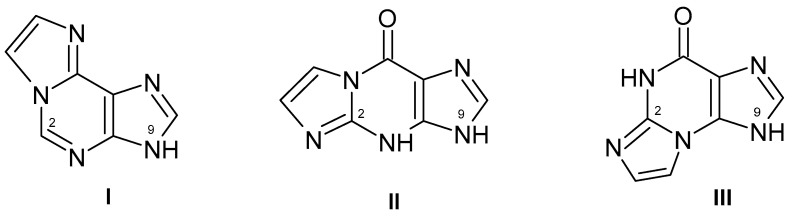
Tricyclic analogs of adenine: 1,N^6^-ethenoadenine (**I**) and guanine (two isomers: 1,N^2^-ethenoguanine (**II**) and N^2^,3-ethenoguanine (**III**)).

**Figure 3 biomolecules-14-00701-f003:**
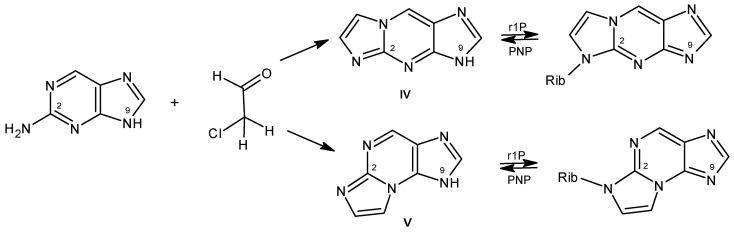
Generation of tricyclic analogs of 2-aminopurine: 1,N^2^-etheno-2-aminopurine (**IV**) and N^2^,3-etheno-2-aminopurine (**V**), and their ribosylation by PNP (Rib = β-D-ribosyl).

**Table 1 biomolecules-14-00701-t001:** Fluorescence parameters of selected 8-azapurine ribosides, compared to other isomorphic purine and purine nucleoside analogs. Unless otherwise indicated, all data refer to neutral species of the fluorophores. Data from references [15,49,53].

Compound	Excitation: λ_max_ [nm]	Emission: λ_max_ [nm]	ϕ	τ [ns]
2-aminopurine riboside	305	380	0.68	~8
8-azaxanthosine *	290	440	>0.1	nd
8-azaguanosine (neutral)	260	347	<0.01	
8-azaguanosine (anionic, pH 10)	278	362	0.55	5.6
8-azaguanine-N7-β-D-riboside	290	420	0.03	nd
8-azaadenosine	278	352	0.068	0.8
8-azainosine (anionic, pH 10)	275	357	0.018	nd
2,6-diamino-8-azapurine- N9-β-D-riboside	285	365	0.9	~6
2,6-diamino-8-azapurine- N7-β-D-riboside	314	420	0.063	1.5;0.45
2,6-diamino-8-azapurine- N8-β-D-riboside	313	430	0.41	10.5
8-aza-7-deaza-isoguanine	288	370	nd	nd
8-aza-isoguanosine (anionic, pH 11)	285	370	~0.1	nd
8-aza-7-deaza-isoguanosine *	285	~430	nd	nd

* The ribosylation site is uncertain; nd = no data

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
