# Peer review of "Chemo-Enzymatic Generation of Highly Fluorescent Nucleoside Analogs Using Purine-Nucleoside Phosphorylase"

_biomolecules, 2024, doi:10.3390/biom14060701_

Round 1
Reviewer 1 Report
Comments and Suggestions for Authors
Comments
The manuscript discusses recent research on the chemical-enzymatic synthesis of highly fluorescent ribosides of nucleobase analogs. It also discusses the synthesis and properties of fluorescent derivatives of purines, such as 8-azapurine and etheno-purine ribosides, obtained using different types of purine nucleoside phosphorylases (PNPs). Interestingly, in some cases, the ribosylation sites differ from the canonical purine N9 atom, and some of the resulting ribosides show fluorescence yield close to 100%.
The manuscript may be of interest to a wide range of readers. I can recommend this manuscript for publication.
Despite the fact that the article is written in good English, it contains some inaccuracies, typos and difficult-to-read sentences. In particular, I would like to recommend the author(s) to rewrite the abstract, if possible. It was hard to read.
I would recommend that the authors, if possible, include more images in the article. This would help to improve understanding.
There is no description provided to Figure 1 in the text, and description to Figure 2 can be found in the text below the Figure itself, which can cause confusion for the reader.
Typos encountered are listed below.
Line 32: dymamic
Line 107: eхpoxide
Line 108: indutrial
Line 164 and 168: 8-Azainosine.... and 8-Azaxanthine....
Line 172 and 300: E. Coli - E. coli
Line 207, 212, 222, 256, 328: E.coli - Insert space
Line 318: Table2 - Insert space
Line 371: synthyesis
It is not very clear what the linear 1,N2-ethenoguanine is (line 314, etc).
It would be good to decode the coefficients φ and ԏ in Table 1.
References to unpublished data are quite common in the text. If these data are going to be published elsewhere, the authors may want to mention this in order to avoid giving the impression that the article is incomplete.
The words in the titles of articles in references should begin with lowercase letters (see ref 1, 5, 9, etc.), and not with uppercase letters (see ref. 2, 3 etc.).
Author Response
REVIEWER’s comments and replies:
REVIEWER I (reply in blue).
The manuscript discusses recent research on the chemical-enzymatic synthesis of highly fluorescent ribosides of nucleobase analogs. It also discusses the synthesis and properties of fluorescent derivatives of purines, such as 8-azapurine and etheno-purine ribosides, obtained using different types of purine nucleoside phosphorylases (PNPs). Interestingly, in some cases, the ribosylation sites differ from the canonical purine N9 atom, and some of the resulting ribosides show fluorescence yield close to 100%.
The manuscript may be of interest to a wide range of readers. I can recommend this manuscript for publication.
Despite the fact that the article is written in good English, it contains some inaccuracies, typos and difficult-to-read sentences. In particular, I would like to recommend the author(s) to rewrite the abstract, if possible. It was hard to read.
- The abstract has been re-written.
I would recommend that the authors, if possible, include more images in the article. This would help to improve understanding.
- We have introduced Figure 3, showing ribosylation scheme for etheno-2-aminopurine derivatives, and corrected Figures 1 and 2. Added is Appendix with Figure A1.
There is no description provided to Figure 1 in the text, and description to Figure 2 can be found in the text below the Figure itself, which can cause confusion for the reader. – This is corrected
Typos encountered are listed below.
Line 32: dymamic -done
Line 107: eхpoxide -done
Line 108: indutrial -done
Line 164 and 168: 8-Azainosine.... and 8-Azaxanthine.... -done
Line 172 and 300: E. Coli - E. coli -done
Line 207, 212, 222, 256, 328: E.coli - Insert space -done
Line 318: Table2 - Insert space -done
Line 371: synthyesis -done
It is not very clear what the linear 1,N2-ethenoguanine is (line 314, etc). –done. “linear” is deleted.
It would be good to decode the coefficients φ and ԏ in Table 1. corrected.
References to unpublished data are quite common in the text. If these data are going to be published elsewhere, the authors may want to mention this in order to avoid giving the impression that the article is incomplete. – We decided to present some of the unpublished data in the Appendix. But its is rather difficult to predict (especially for the 73-yers old man like me) which data will be published and when.
The words in the titles of articles in references should begin with lowercase letters (see ref 1, 5, 9, etc.), and not with uppercase letters (see ref. 2, 3 etc.). OK, this is corrected.
Reviewer 2 Report
Comments and Suggestions for Authors
The manuscript entitled "Chemo-enzymatic generation of highly fluorescent nucleoside analogs using purine-nucleoside phosphorylase – a mini-review” was submitted for its publication in Biomolecules
In this work, the authors review the application of PNP enzymes from different sources to the preparation of fluorescent ribosides of nucleobase analogues, paying special attention to aza and deaza compounds. The paper compiles a lot of information on the subject, and provides an overview of the fluorescent characteristics of the usual and unusual nucleoside obtained. Finally, it comments on the potential of this compounds in biophysical and analytical research and gives a very accurate opinion, from my point of view, of the potential of synthetic biology to design enzymes for specific substrates.
The review fits well with the field of Biotechnological and Biomedical Applications of Enzymes Involved in the Synthesis of Nucleosides and Nucleotides—Volume II, special issue. Therefore, I recommend its publication in this journal.
However, the following comments could be considered to improve the manuscript before publication.
- Table 2 should be referenced in section 4.1
- For clarity in the discussion, the numbers of all atoms in the structures of Figure 2 should be indicated. Perhaps it would be appropriate to use regular font when the number of the N corresponds to its position in the cycle (for example N9) and superscript when referring to an N bonded to carbon (for example N6)
- It would be a great contribution to the work to include a discussion about unexpected phosphorylations of PNP with unusual substrates.
Author Response
REVIEWER II (reply in violet).
The manuscript entitled "Chemo-enzymatic generation of highly fluorescent nucleoside analogs using purine-nucleoside phosphorylase – a mini-review” was submitted for its publication in Biomolecules
In this work, the authors review the application of PNP enzymes from different sources to the preparation of fluorescent ribosides of nucleobase analogues, paying special attention to aza and deaza compounds. The paper compiles a lot of information on the subject, and provides an overview of the fluorescent characteristics of the usual and unusual nucleoside obtained. Finally, it comments on the potential of this compounds in biophysical and analytical research and gives a very accurate opinion, from my point of view, of the potential of synthetic biology to design enzymes for specific substrates.
The review fits well with the field of Biotechnological and Biomedical Applications of Enzymes Involved in the Synthesis of Nucleosides and Nucleotides—Volume II, special issue. Therefore, I recommend its publication in this journal.
However, the following comments could be considered to improve the manuscript before publication.
- Table 2 should be referenced in section 4.1 – It is now referenced in sections 4.1, 4.2 and 4.3.
- For clarity in the discussion, the numbers of all atoms in the structures of Figure 2 should be indicated. Perhaps it would be appropriate to use regular font when the number of the N corresponds to its position in the cycle (for example N9) and superscript when referring to an N bonded to carbon (for example N6) – We are aware of this convention, and corrected the text, but leave to the Editor a decision of its actual implementation.
- It would be a great contribution to the work to include a discussion about unexpected phosphorylations of PNP with unusual substrates. – I do not feel fully competent in this matter, but the brief discussion of this phenomenon is included in section 3.3 and in the “Conclusions”. There is also an Appendix added to illustrate some kinetic peculiarities observed by us.
Reviewer 3 Report
Comments and Suggestions for Authors
The manuscript of Wierzchowki et al. as stated by the title gives a brief overview on the chemo-enzymatic generation of highly fluorescent nucleoside analogs using purine-nucleoside phosphorylase, and therefore focuses on purines and purine isosters and their enzymatic glycosylation, as well on their fluorescent properties and some potential uses for specific members. The paper therefore perfectly fits the targeted special issue on “Biotechnological and Biomedical Applications of Enzymes Involved in the Synthesis of Nucleosides and Nucleotides”.
Overall the manuscript is worthwhile publishing in this special issue provided a thorough spelling update is made (numerous typo’s!) and some clear structural figures are included to enlighten the manuscript. Some further mistakes are indicated in the more detailed list.
Some more detailed comments:
General remark: Overall, some figures should be included, depicting the most important fluorescent structures as well as the enzymatic reaction for synthesizing them.
Line 12: although it is clear the enzymatic synthesis provides ribofuranose derivatives, I would clearly state this, at least at first appearance. Therefore correct line 12 to “ribofuranosides of nucleobase analogs”. In analogy, the substrate r1P (line 15) however is generally known to be the furanose derivative, so the actual phrasing is fine (although chemically speaking a pyranose would be possible as well).
Line 55-56: both line 55 and line 56 should read "isothiazolo-pyrimidine". With the thiazolopyrimidine no C-nucleoside could be constructed.
line 65: correct text to improve reading to "offers the possibility"
Line 69: correct to "linked to an organic ..."
Line 92: correct to "and other good ribose donors..."
Line 95: "thymidine phosphorylase"
Line 105: "stability"
Line 108: "recently"
Line 123: correct to “other enzymatic systems” (or “another enzymatic system”).
Line 124: correct to "syntheses". May I request the authors to carefully check the complete manuscript for ubiquitous obvious spelling mistakes.
Line 128: in view of ref 49 of the same author, being less easy to consult, this referee has difficulties to evaluate the novelty of the present review compared to previous one. But it therefore is considered positive to have this new submission now for an open access source.
Figure 1: the figures are a bit awkward with unusually long bond lengths for substituents. Please adapt accordingly.
Line 172: use small caps for "coli".
Lines 171-176: for readability use at least two sentences.
Line 193: it is unclear what the "N9-phosphonomethoxy analog of 8-azaguanine" could be? The authors probably mean a "N9-phosphonomethoxyalkyl" analog? (recurring “chemical mistake”, e.g. at line 227, being the “phosphonomethoxyethyl” analog).
Table 1: when the table states the data are for the anionic species, preferentially the pH should be added at which the analysis was made.
Just another few examples of sloppy typo’s: “revealing” (246), inhibitor (248), purine (253).
Line 259-260: I understand what the authors mean, but the phrasing is a bit awkward as it sounds now whether 2,6-diaminopurine and 4,6-diaminopyrazolo[3,4-d]pyrimidine are the same, quod non. Please rephrase.
Line 274: correct to "is probably a poor or no substrate".
Line 279 (and repeatedly further): no hyphen needed for "tricyclic".
Figure 2: adjust the structure sizes, provide a number and place them in one single row. Use the respective numbers in the text for non-chemical readers.
The reference list overall is fine, but please carefully check all references again as some minor faults were noticed upon quick evaluation. Some examples: the use of a different pitch (especially for DOI numbers), not all year numbers are in bold, wrong DOI number as in ref 31, some missing author names as in ref 73.
Comments on the Quality of English LanguageEspecially detailed spelling check required; the English phrasing is understandable.
Author Response
REVIEWER III (reply in green).
The manuscript of Wierzchowki et al. as stated by the title gives a brief overview on the chemo-enzymatic generation of highly fluorescent nucleoside analogs using purine-nucleoside phosphorylase, and therefore focuses on purines and purine isosters and their enzymatic glycosylation, as well on their fluorescent properties and some potential uses for specific members. The paper therefore perfectly fits the targeted special issue on “Biotechnological and Biomedical Applications of Enzymes Involved in the Synthesis of Nucleosides and Nucleotides”.
Overall the manuscript is worthwhile publishing in this special issue provided a thorough spelling update is made (numerous typo’s!) and some clear structural figures are included to enlighten the manuscript. Some further mistakes are indicated in the more detailed list. – We have checked the text for (many) typos. Figure 3 and Appendix are added to clarify the discussion.
Some more detailed comments:
General remark: Overall, some figures should be included, depicting the most important fluorescent structures as well as the enzymatic reaction for synthesizing them. - We have introduced Figure 3, showing ribiosylation scheme for the etheno-2-aminopurine derivatives.
Line 12: although it is clear the enzymatic synthesis provides ribofuranose derivatives, I would clearly state this, at least at first appearance. Therefore correct line 12 to “ribofuranosides of nucleobase analogs”. In analogy, the substrate r1P (line 15) however is generally known to be the furanose derivative, so the actual phrasing is fine (although chemically speaking a pyranose would be possible as well). – this has been done
Line 55-56: both line 55 and line 56 should read "isothiazolo-pyrimidine". With the thiazolopyrimidine no C-nucleoside could be constructed. -done
line 65: correct text to improve reading to "offers the possibility" -done
Line 69: correct to "linked to an organic ..." -done
Line 92: correct to "and other good ribose donors..." -done
Line 95: "thymidine phosphorylase" -done
Line 105: "stability" -done
Line 108: "recently" -done
Line 123: correct to “other enzymatic systems” (or “another enzymatic system”). -done
Line 124: correct to "syntheses". May I request the authors to carefully check the complete manuscript for ubiquitous obvious spelling mistakes. –this is done.
Line 128: in view of ref 49 of the same author, being less easy to consult, this referee has difficulties to evaluate the novelty of the present review compared to previous one. But it therefore is considered positive to have this new submission now for an open access source. – Reference 49 is dated 2017, and there are several new papers to summarize and discuss (refs. 57b, 66, 87, 90, 92 in particular).
Figure 1: the figures are a bit awkward with unusually long bond lengths for substituents. Please adapt accordingly. This is corrected.
Line 172: use small caps for "coli". -done
Lines 171-176: for readability use at least two sentences. -done
Line 193: it is unclear what the "N9-phosphonomethoxy analog of 8-azaguanine" could be? The authors probably mean a "N9-phosphonomethoxyalkyl" analog? (recurring “chemical mistake”, e.g. at line 227, being the “phosphonomethoxyethyl” analog). -done
Table 1: when the table states the data are for the anionic species, preferentially the pH should be added at which the analysis was made. OK. The pH values are added.
Just another few examples of sloppy typo’s: “revealing” (246), inhibitor (248), purine (253). -corrected
Line 259-260: I understand what the authors mean, but the phrasing is a bit awkward as it sounds now whether 2,6-diaminopurine and 4,6-diaminopyrazolo[3,4-d]pyrimidine are the same, quod non. Please rephrase. - This has been rephrased.
Line 274: correct to "is probably a poor or no substrate". -done
Line 279 (and repeatedly further): no hyphen needed for "tricyclic". -corrected
Figure 2: adjust the structure sizes, provide a number and place them in one single row. Use the respective numbers in the text for non-chemical readers.
The reference list overall is fine, but please carefully check all references again as some minor faults were noticed upon quick evaluation. Some examples: the use of a different pitch (especially for DOI numbers), not all year numbers are in bold, wrong DOI number as in ref 31, some missing author names as in ref 73. The Reference section is corrected.
Comments on the Quality of English Language
Especially detailed spelling check required; the English phrasing is understandable. This has been done.
Submission Date
10 May 2024
Date of this review
17 May 2024 18:07:55
The authors thank the Referee for valuable comments and apologize for all the errors and mistakes.
Round 2
Reviewer 1 Report
Comments and Suggestions for Authors
Table 2, column Compound: " ...Riboside" must be written with a lowercase (small) letter, ...-riboside
Author Response
The compound names in Table 2 have been corrected according to Reviewer's suggestion.
Reviewer 2 Report
Comments and Suggestions for Authors
The manuscript entitled "Chemo-enzymatic generation of highly fluorescent nucleoside analogs using purine-nucleoside phosphorylase – a mini-review” has been modified taking into account the suggestions made by the reviewers. This version has been improved enough to be published in Biomolecules, Biotechnological and Biomedical Applications of Enzymes Involved in the Synthesis of Nucleosides and Nucleotides—Volume II, special issue.
Author Response
Thank you for reviewing again. There is only minor correction in Table 2, no other changes.
Reviewer 3 Report
Comments and Suggestions for Authors
The manuscript of Wierzchowki et al. has been thoroughly revised along the lines suggested by the referees and now conforms to the quality required for publication in Biomolecules.
This referee has only 2 further remarks:
The new Figure 3 depicts the (concise) synthesis of the tricyclic analogs of purines, while a more mechanistic scheme to obtain the tricyclic derivatives was expected to meet our request, but this is fine. However chloroacetic acid is depicted in the scheme as the required reagent, which I do not believe will work.
Second: normally a revised manuscript is always accompanied with a thorough rebuttal explaining all changes. We did not receive such rebuttal which is quite unusual!
Author Response
This referee has only 2 further remarks:
The new Figure 3 depicts the (concise) synthesis of the tricyclic analogs of purines, while a more mechanistic scheme to obtain the tricyclic derivatives was expected to meet our request, but this is fine. However chloroacetic acid is depicted in the scheme as the required reagent, which I do not believe will work.
- The chemistry of chloroacetaldehyde reaction with purines and purine nucleosides is discussed in details in reference [86] by Boryski’s group from Poznan. We think that repeating the detailed schemes of the reaction is unnecessary. Virta et al. (ref.[91]) described reaction of 2-aminopurine-9-riboside with aqueous chloroacetaldehyde, resulting in 1,N2-etheno- derivative. This derivative is spectrally nearly identical with our product IV, except some minor difference in fluorescence yield. We have repated this reaction, searching for possible second, more fluorescent isomer, but its yield of formation was very low, if any. The analogous reaction of 2-aminopurine base was somewhat complicated by low solubility of the substrate, but this was overcame by addition of acetic acid plus aliquot of HCl, and pH of the reaction was regulated by sodium bicarbonate addition. The reaction took ~48 hrs at room temperature. Two products were clearly seen on HPLC profile with fluorescence detection. Still, the yield of V was less than 5%. To our surprise, this minor product, purified by semi-prep HPLC, was rapidly ribosylated by PNP, giving non-typical riboside, identified using NMR, as described in ref. [92]. Both products, IV and V, were somewhat unstable due to slow air oxidation, but this did not interfere with spectroscopic or kinetic experiments. We think these informations are sufficient for the Referee, if he (or she) thinks it is worth repeating, but we do not intend to repeat them in the text.
Second: normally a revised manuscript is always accompanied with a thorough rebuttal explaining all changes. We did not receive such rebuttal which is quite unusual!
- We always regard critical reviews as occasions to improve the quality of our manuscripts. So we have introduced many changes – including some modifications which were not directly required by the Referees, but in our view were necessary because of the recently published papers related to this subject. All the changes were marked in the text by color codes – changes suggested by this Referee were marked green. Additionally, the Editor suggested some changes in references (eliminate excessive self-citations). We have added some new sentences, as well as the Appendix, to the discussion (all marked in red). But all the points of the review have been answered (sometimes by one word “done”).
Submission Date
10 May 2024
Date of this review
03 Jun 2024 11:32:59
Date of the reply – 05 Jun 2024, 15:06.